# Retesting Schemes That Improve Test Quality and Yield Using a Test Guardband

Chung-Huang Yeh [1,*] and Jwu-E Chen [2]

1   Intelligent Manufacturing Engineering, Minth University, Hsinchu 307, Taiwan
2   Department of Electrical Engineering, National Central University (NCU), Taoyuan 300, Taiwan; jwu.e.chen@gmail.com
*   Correspondence: yehsony@gmail.com

**Abstract:** The digital integrated circuit (IC) testing model module is applied in this study to simulate the fabrication and testing of integrated circuits. The yield and quality of ICs are analyzed by assuming that the wafer devices under test conditions are normal probability distributions. The difficulties of testing and verification become increasingly great as the design function of the chip becomes remarkably complex. Conversely, the automotive industry chip supply chain has been substantially affected since the COVID-19 outbreak. The shortage of chips in the auto-market has always existed; therefore, increasing available chips under a limited production capacity has become a top priority. Therefore, this study applies the digital integrated circuit testing model (DITM) and proposes a retest plan. This method does not require considerable time to collect large wafer data, nor does it require additional hardware equipment. Furthermore, the required test quality parameters are set, and the test is repeated on the device by adjusting the test guardband (TGB). Moreover, three retesting schemes are proposed to improve the IC test quality ($Y_q$) and test yield ($Y_t$) to meet the requirements of consumers for product quality. A set of 2021 IEEE International Roadmap for Devices and Systems (IRDS) parameters is used to demonstrate the three proposed retesting schemes. The simulation results from the 2021 IRDS data prove that the retest method can effectively improve the test yield ($Y_t$). A comparison of the estimated results of the three retest methods shows that using the repeat test method can maximize the test yield without sacrificing the test quality ($Y_q$). By contrast, repeat testing can indeed improve the test yield ($Y_t$) by 14% or more. Moreover, the increase in sellable ICs not only increases additional earnings for corporations, but also alleviates the current global shortage of automotive ICs.

**Keywords:** guardband test; defect level; test specification; zero defect; test quality





## 1. Introduction

The process of integrated circuits (ICs) has rapidly developed and progressed from the initial 90 to 1 nm. In addition, a large wafer indicates a small line width, becoming increasingly complicated relative to the process. The chips produced by advanced processes have additional functions; thus, their complexity also becomes increasingly high [1–4]. Therefore, the effective verification and testing of ICs has become an important issue in academic and industrial circles. Moreover, the development of testing technology is different from that of semiconductor device fabrication [5,6], and the inaccuracy of automatic test equipment (ATE, IC tester) leads to increasing yield losses. Furthermore, using an ATE (IC tester) with insufficient testing capabilities to create high-quality products in the future will be remarkably challenging. This thesis quantifies the process of testing and semiconductor chip manufacturing. Assuming that the characteristics of chip products are normally distributed, the DITM (integrated circuit testing model) model [7] is used to estimate the test quality ($Y_q$) and yield ($Y_t$) of chip products. The progress rate in future manufacturing industries is unpredictable. Therefore, the IC test model (DITM) is utilized to estimate the

distribution trend of future chip yields using the electrical characteristics of existing products and current manufacturing technology. The semiconductor manufacturing company's greatest goal is to produce high-quality (zero-defect) chip products [8–11]. The quality requirements are strict, especially in the aviation and medical electronics industries, with high safety requirements. Semiconductor quality standards are typically expressed in terms of defects per million. However, the defect index standard for key parts of automobiles and aviation has been increased to a few parts per billion. The current testing technology and IC tester (ATE) capabilities cannot meet the quality ($Y_q$) and yield ($Y_t$) requirements of the chip; thus, the testing house must find a more effective alternative testing method [12–18] than the current one. For example, S.C. Horng proposed a two-stage method [12] based on the ordinal optimization theory to achieve less overkills and retests and applied it to semiconductor products to reduce overkills at a tolerable retesting rate. In addition, Sisir Kumar Jena identified these acceptable circuits (AcICs) by retesting [13], thereby indirectly improving the effective yield. Circuits may produce incorrect but ignorable results for some test patterns; thus, testing is continued until all test patterns are applied. If the level of deviation has a minor effect on the overall performance of the circuit, the circuit may be accepted as a passing IC. This method can greatly improve the yield by retesting.

At present, due to the slow technological progress of ATE (IC testers), testing capabilities lag far behind semiconductor manufacturing technology [19]. The testing yield of chips will reduce in the future due to the tester's inaccuracy [5,6]. Put simply, the reduction in the yield rate may result in a shortage of chips and even lead to the disconnection of the semiconductor supply chain. A structural shortage has been observed in the past two years due to the impact of the epidemic on the chip industry chain. Coupled with the high chip demand for new energy vehicles, the global chip shortage has caused a supply crisis in the global semiconductor industry chain (global chip shortage). Three effective retesting schemes [20–22] are proposed to improve the test quality ($Y_q$) and yield ($Y_t$) to solve the aforementioned problem. The goal of zero defects is achieved through the retesting of the changes in the test guardband (TGB). A set of 2021 IRDS [23] parameters is used to demonstrate and cooperate with the IC test model (DITM) to estimate future chip yield trends. The retesting scheme is also employed to reduce the occurrence of missing errors (β) and killing errors (α) to improve the quality ($Y_q$) and yield ($Y_t$). The three retesting schemes are incorporated into the IRDS table after the abovementioned simulation estimation and comparison are performed. The comparison results indicate that retesting not only enhances the chip test yield ($Y_t$), but also improves the performance of ATE test equipment. In addition, the retesting scheme not only increases the sales of additional zero-defect chips, but also alleviates the severe shortage of automotive chips worldwide.

## 2. Semiconductor Manufacturing and Testing Process

As Figure 1 shows the process of semiconductor IC development and manufacturing and its delivery to the testing house. The circuit is developed and verified by the design house and then sent to the foundry for wafer production and manufacturing. Then, the IC is sent to the chip testing house for verification and testing.

Suppose N number of chips is manufactured in the process of IC manufacturing (Figure 1), The chips manufactured by the wafer foundry can be divided into bad parts (B) and good parts (G) according to the established design specifications (DS, Product Design Specifications of Semiconductor Integrated Circuits). The manufacturing yield ($Y_m$) after manufacturing can be expressed by the following formula: $Y_m = G/N$. The produced chips are then sent to IC testing houses for testing. These chips can be classified into failed (F) and passed (P) parts judging by the test specification (TS, Product Test Specifications of Semiconductor Integrated Circuits) provided by the manufacturer. If the test process is perfect, then the test yield ($Y_t$) is equal to the manufacturing yield ($Y_m$). However, the test cannot be perfect, and a test error is caused by the errors of the ATE or the test method. Killing errors (type-I error (α), number of good chips that fail rigorous testing) and missing errors (type-II error (β), number of bad chips that pass rigorous testing) may also exist.

Missing errors result in the return of products, which seriously affects the company's image. Killing errors result in a loss of product yield, thus reducing the revenue and profit.

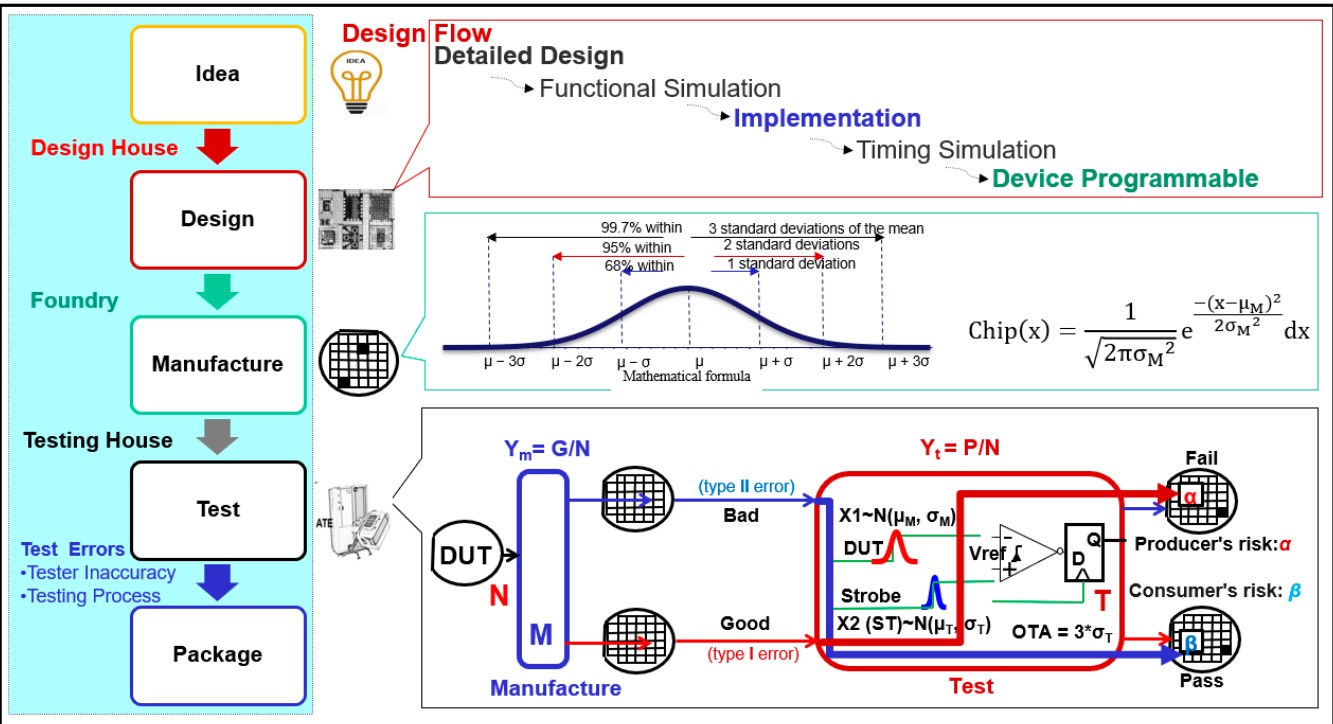

**Figure 1.** Chip development and testing procedures.

### 2.1. Semiconductor Manufacturing Yield Calculation

The normal distribution is a theoretical model that can provide relatively accurate descriptions and inferences for the obtained data from the empirical research with the mean parameter μ and the standard deviation parameter σ. The normal distribution type is often used in the general traditional statistical analysis and estimation owing to its computational accuracy. The mean, μ, and standard deviation, σ, of the random variable, X, of the probability density function of the normal distribution can be expressed as N(x; μ, σ). Generally speaking, the probability density function in the mathematical form of the normal distribution can also be determined as follows:

$$f(x) = \frac{1}{\sqrt{2\pi}\sigma} e^{-\frac{1}{2}\left(\frac{X-\mu}{\sigma}\right)^2} dx \tag{1}$$

Semiconductor manufacturing steps include the following: deposition, photoresist, lithography, etching, ionization, and packaging thousands of tedious manufacturing processes. After the chip is manufactured at the wafer foundry, the electrical characteristic of the DUT (device under test) may present a normal probability distribution instead of a fixed value due to the changes in the semiconductor device fabrication and uncertainty of the manufacturing process. Herein, the delay time of the DUT was assumed to be normally distributed. The standard deviation, $\sigma_M$, and mean value, $\mu_M$, of the electrical characteristics of a semiconductor product can be expressed as Chip (x) = N (x; $\mu_M$, $\sigma_M$):

$$\begin{aligned} Y_m &= Manufacturing\ Yield\ (\%) \\ &= \int_{-\infty}^{DS} Chip(x)dx \\ &= \int_{-\infty}^{DS} \frac{1}{\sigma_M \sqrt{2\pi}} e^{-\frac{1}{2}\left(\frac{X-\mu_M}{\sigma_M}\right)^2} dx \\ &= \int_{-\infty}^{\frac{DS-\mu_M}{\sigma_M}} \frac{1}{\sqrt{2\pi}} e^{-\frac{1}{2}(x)^2} dx. \end{aligned} \tag{2}$$

Figure 2 represents our design of a CPU chip. The following values are the electrical characteristics of the DUT (chip) that can present a normal probability distribution instead of a fixed value due to the changes in the semiconductor device fabrication and uncertainty of the manufacturing process of the chip: the DS is 0.858 GHz (DS = 1165 ps), the standard deviation value is $\sigma_M$ = 100 ps, and the mean value is $\mu_M$ = 1000 ps. Chip X ~ N (x; $\mu_M$ = 1000 ps and $\sigma_M$ = 100 ps) can be used to represent the chip distribution. The distribution of the chip delay time is shown in Figure 2, where the vertical and horizontal axes represent the probability density and time parameter of the circuit characteristics, respectively. The manufacturing yield (95% ($Y_m$ = P[X < DS] = P[Good])) can be obtained in accordance with the derivation and calculation of Formula (1).

$$Y_m = \int_{-\infty}^{DS} \frac{1}{\sigma_M \sqrt{2\pi}} e^{-\frac{1}{2}\left(\frac{X-\mu_M}{\sigma_M}\right)^2} dx = \int_{-\infty}^{\frac{DS-\mu_M}{\sigma_M}} \frac{1}{\sqrt{2\pi}} e^{-\frac{1}{2}(x)^2} dx$$
$$= \int_{-\infty}^{1165} \frac{1}{\sigma_M \sqrt{2\pi}} e^{-\frac{1}{2}\left(\frac{X-1000}{100}\right)^2} dx = \int_{-\infty}^{\frac{1165-1000}{100}} \frac{1}{\sqrt{2\pi}} e^{-\frac{1}{2}(x)^2} dx = 95\%.$$

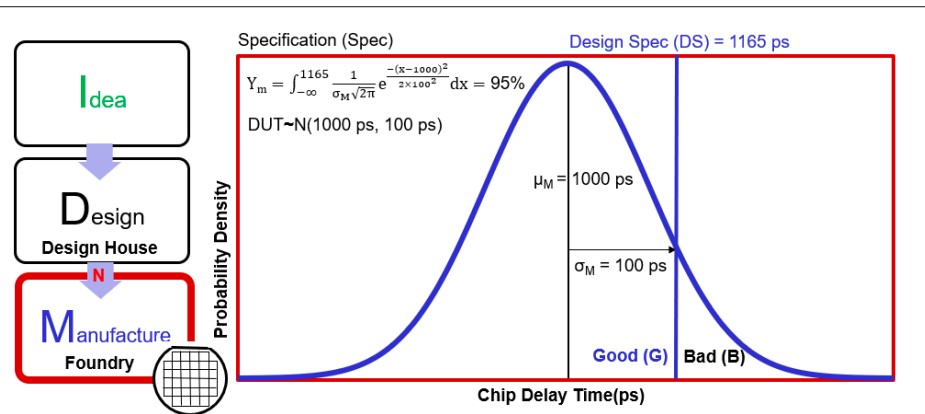

**Figure 2.** Estimation of the manufacturing yield of a wafer.

### 2.2. Threshold Test System for Determining the Chip Quality

The test aimed to use the comparison of the DS and TS to distinguish whether the DUT met the design requirements. Testing is also a sorting activity; it involves selecting products that meet the design and TS standards from production wafers. On the contrary, the defective parts that do not meet the DS requirements are disposed of to avoid customer returns. The circuit design function of modern chips is remarkably powerful, and the parameter relationship is also quite complicated. The test items of the chip include the following: functional, parameter, and delay tests. The abovementioned different test parameters and the capability development of the IC tester (ATE) are considered, and the accuracy and convenience of the test results are measured during predictions. Therefore, this paper determined whether the IC was bad or good by comparing the chip delay time and the strobe timing of the tester.

Figure 3 shows the threshold test model [7], where X2 (ST) is the ATE signal (strobe) and X1 is the chip delay time of the DUT. In an ATE, the tester sends a strobe signal to compare the timing (X1, X2) and product's response ("passed" or "failed"). When the X1 signa arrives faster (X1 < X2) than the X2 signal sent by the tester, the chip is then classified into a qualified part and the pass signal is sent by the ATE. Conversely, when the X1 signa of the chip arrives slower (X1 > X2) than the X2 signal from the tester, the ATE sends a failure signal and classifies the IC chip as a failed part.

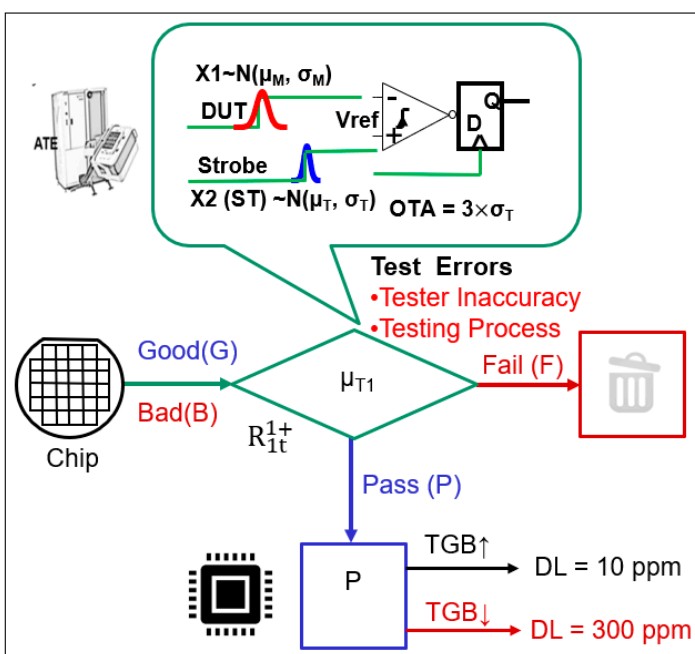

**Figure 3.** Traditional testing methods that are only performed once.

*2.3. Chip Test Yield Estimation ($Y_t$)*

Wafers that do not meet the product specifications result from uncertainty and error factors occurring during the foundry's manufacturing process (lithography, etching, and deposition). Therefore, defective chips can be identified through testing procedures and other mechanisms. The signal sent by the ATE (IC tester) can suffer from inaccuracy and edge displacement in the testing process. Therefore, the test capabilities of the ATE (IC tester) were assumed to be distributed normally in this paper. The parameter of the electrical distribution of the ATE (IC tester) can be expressed as $X \sim N(x; \mu_T, \sigma_T)$, where $\mu_T$ (tester) is the average value and $\sigma_T$ (tester) is the standard deviation. According to the distribution calculation of the ATE (IC tester), the yield ($Y_t$) of the test can be expressed as $Y_t = P[X < Y] = P[\text{pass}]$, and the calculation process is as follows:

$$
\begin{aligned}
R_{1t}^{1+} \text{Test Yield} &= Y_t \\
&= \int_{-\infty}^{\infty} \int_x^{\infty} \text{Chip}(x) \, \& \, \text{Tester}(x,y) dy dx \\
&= \int_{-\infty}^{\infty} \text{Chip}(x) \int_x^{\infty} \text{Tester}(y) dy dx \\
&= \int_{-\infty}^{\infty} \frac{1}{\sigma_M \sqrt{2\pi}} e^{-\frac{1}{2}\left(\frac{x-\mu_M}{\sigma_M}\right)^2} \int_x^{\infty} \frac{1}{\sigma_T \sqrt{2\pi}} e^{-\frac{1}{2}\left(\frac{y-\mu_T}{\sigma_T}\right)^2} dy dx \\
&= \int_{-\infty}^{\infty} \frac{1}{\sqrt{2\pi}} e^{-\frac{1}{2}(x)^2} \int_{\frac{\mu_M + \sigma_M x - \mu_T}{\sigma_T}}^{\infty} \frac{1}{\sqrt{2\pi}} e^{-\frac{1}{2}y^2} dy dx.
\end{aligned}
\tag{3}
$$

The traditional test method (Figure 3) that only tests the DUT once is represented by $R_{1t}^{1+}$.

In the process of wafer testing, the mechanical and electrical parameters of the tester affect the test results; the way engineers use these parameters and the operation of the tester also indirectly influence the test results. Testing cannot be perfect; thus, testing errors will continue to emerge. Therefore, in addition to the yield ($Y_t$) of the factor, the test quality ($Y_q$) has to be considered in the test results.

The quality of semiconductor products Is often expressed by the defect level (DL), which is the ratio of salable to defective IC products. The DL unit is usually expressed in ppm (parts per million): DL = P[Bad | Pass] = P[(X > DS) ∩ (X < ST)] / P[X < ST]. Then, we used 10 ppm as an example; 10 ppm suggested that the foundry produced 1 million chip products, of which 10 chips may could have defects. From the aspects of use, production, and price, the quality acceptable to OEM manufacturers and consumers should be between

DL = 200–300 ppm. A DL with a low number represents a high-quality product with a low return rate, while that with a high number represents a low-quality product with a high return rate.

$$\text{DL(Defect Level)} = \frac{P[\text{Bad}|\text{Pass}]}{Y_t}$$

$$= \frac{\text{Missing Errors}}{Y_t}$$

$$= \frac{\int_{DS}^{\infty} \int_{x}^{\infty} \text{Chip(x) \& Tester(x,y)} dy dx}{\int_{-\infty}^{\infty} \int_{x}^{\infty} \text{Chip(x) \& Tester(x,y)} dy dx}$$

$$= \frac{\int_{DS}^{\infty} \text{Chip(x)} \int_{x}^{\infty} \text{Tester(y)} dy dx}{\int_{-\infty}^{\infty} \text{Chip(x)} \int_{x}^{\infty} \text{Tester(y)} dy dx}$$

$$= \frac{\int_{DS}^{\infty} \frac{1}{\sigma_M \sqrt{2\pi}} e^{\frac{-(x-\mu_M)^2}{2\sigma_M^2}} \int_{x}^{\infty} \frac{1}{\sigma_T \sqrt{2\pi}} e^{\frac{-(y-\mu_T)^2}{2\sigma_T^2}} dy dx}{\int_{-\infty}^{\infty} \frac{1}{\sigma_M \sqrt{2\pi}} e^{\frac{-(x-\mu_M)^2}{2\sigma_M^2}} \int_{x}^{\infty} \frac{1}{\sigma_T \sqrt{2\pi}} e^{\frac{-(y-\mu_T)^2}{2\sigma_T^2}} dy dx} \tag{4}$$

$$= \frac{\int_{\frac{DS-\mu_M}{\sigma_M}}^{\infty} \frac{1}{\sqrt{2\pi}} e^{-\frac{1}{2}(x)^2} \int_{\frac{\mu_M+\sigma_M x-\mu_T}{\sigma_T}}^{\infty} \frac{1}{\sqrt{2\pi}} e^{-\frac{1}{2}y^2} dy dx}{\int_{-\infty}^{\infty} \frac{1}{\sqrt{2\pi}} e^{-\frac{1}{2}(x)^2} \int_{\frac{\mu_M+\sigma_M x-\mu_T}{\sigma_T}}^{\infty} \frac{1}{\sqrt{2\pi}} e^{-\frac{1}{2}y^2} dy dx}$$

$$\text{Missing Errors} = [\text{Bad}|\text{Pass}]$$
$$= \int_{DS}^{\infty} \int_{x}^{\infty} \text{Chip(x) \& Tester(x,y)} dy dx$$
$$= \int_{DS}^{\infty} \text{Chip(x)} \int_{x}^{\infty} \text{Tester(y)} dy dx$$
$$= \int_{DS}^{\infty} \frac{1}{\sigma_M \sqrt{2\pi}} e^{-\frac{1}{2}\left(\frac{x-\mu_M}{\sigma_M}\right)^2} \int_{x}^{\infty} \frac{1}{\sigma_T \sqrt{2\pi}} e^{-\frac{1}{2}\left(\frac{y-\mu_T}{\sigma_T}\right)^2} dy dx \tag{5}$$
$$= \int_{\frac{DS-\mu_M}{\sigma_M}}^{\infty} \frac{1}{\sqrt{2\pi}} e^{-\frac{1}{2}(x)^2} \int_{\frac{\mu_M+\sigma_M x-\mu_T}{\sigma_T}}^{\infty} \frac{1}{\sqrt{2\pi}} e^{-\frac{1}{2}y^2} dy dx \quad.$$

## 3. Impact of Guardband Testing on Yield

During the testing process of the semiconductors, test inaccuracies occurred (Figure 4) due to the edge placement of the ATE. Therefore, the TGB must be considered to avoid the inaccuracy of the ATE [24]. The distance between the TS and DS is shown in Figure 5; TGB is defined as the difference between the DS and TS (TGB = DS − TS). Changing the TS and expanding the TGB (DS − TS = TGB ↑) increases $Y_t$ and reduces $Y_q$. In contrast, the $Y_q$ of the product declines by reducing the TGB (DS − TS = TGB ↓). Therefore, the choice between the reduction in and expansion of the TGB can change the values of $Y_q$ and $Y_t$ and serve as a reference for measuring $Y_q$ and $Y_t$.

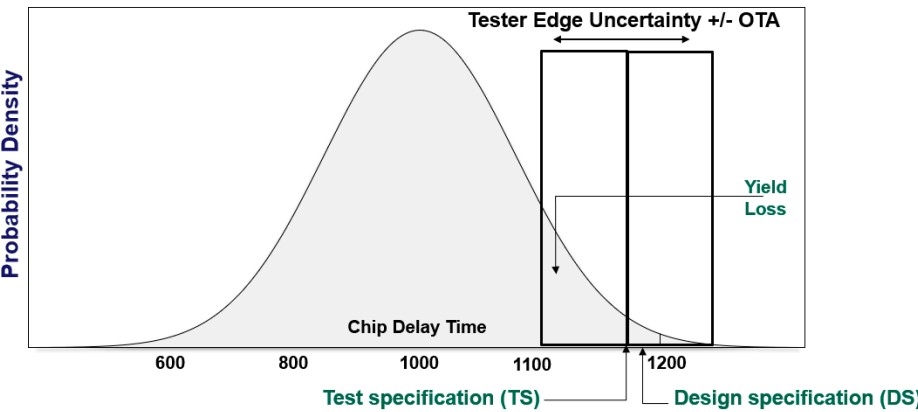

**Figure 4.** Tester edge distribution and losses.

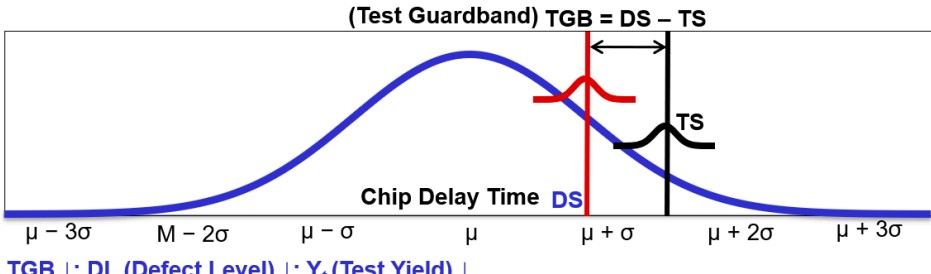

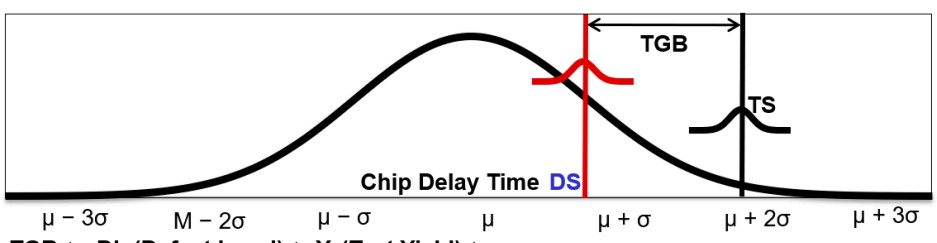

**Figure 5.** Test guardband (TGB) impact test for $Y_t$ and DL.

For instance, the design house develops an advanced chip, the electrical characteristic parameters of the chip can be expressed as $X \sim N$ (x; $\mu_M$ = 1000 ps and $\sigma_M$ = 100 ps), and its DS = 1165 ps. We can obtain $Y_m$ = 95% (manufacturing yield) by substituting the previously estimated Formula (2). Then, the DUT performed with the IC tester (ATE) with a characteristic parameter overall timing accuracy (OTA) = 120 ps (OTA = $\sigma_T \times 3$, $\sigma_T$ = 40 ps) and test quality ($Y_q$) requirement was set to DL = 300 ppm. The test yield = 77.73% ($Y_t$) can be obtained (Figure 6) by using TS ($\mu_T$) = 1082 ps (TGB, 1165 ps − 1082 ps = 83 ps) to assess the DUT.

$$\text{Test Yield}(Y_t)R_{1t}^{1+}$$
$$= \int_{-\infty}^{\infty} \frac{1}{\sigma_M\sqrt{2\pi}} e^{-\frac{1}{2}\left(\frac{X-\mu_M}{\sigma_M}\right)^2} \int_x^{\infty} \frac{1}{\sigma_T\sqrt{2\pi}} e^{-\frac{1}{2}\left(\frac{y-\mu_T}{\sigma_T}\right)^2} dydx$$
$$= \int_{-\infty}^{\infty} \frac{1}{\sigma_M\sqrt{2\pi}} e^{-\frac{1}{2}\left(\frac{X-1000}{100}\right)^2} \int_x^{\infty} \frac{1}{\sigma_T\sqrt{2\pi}} e^{-\frac{1}{2}\left(\frac{y-1082}{40}\right)^2} dydx$$
$$= \int_{-\infty}^{\infty} \frac{1}{\sqrt{2\pi}} e^{-\frac{1}{2}(x)^2} \int_{\frac{\mu_M+\sigma_M x-\mu_T}{\sigma_T}}^{\infty} \frac{1}{\sqrt{2\pi}} e^{-\frac{1}{2}y^2} dydx$$
$$= \int_{-\infty}^{\infty} \frac{1}{\sqrt{2\pi}} e^{-\frac{1}{2}(x)^2} \int_{\frac{1000+100x-1082}{40}}^{\infty} \frac{1}{\sqrt{2\pi}} e^{-\frac{1}{2}y^2} dydx = 77.73\%$$

$$\text{DL}(\text{Defect Level})$$
$$= \frac{\int_{\frac{DS-\mu_M}{\sigma_M}}^{\infty} \frac{1}{\sqrt{2\pi}} e^{-\frac{1}{2}(x)^2} \int_{\frac{\mu_M+\sigma_M x-\mu_T}{\sigma_T}}^{\infty} \frac{1}{\sqrt{2\pi}} e^{-\frac{1}{2}y^2} dydx}{\int_{-\infty}^{\infty} \frac{1}{\sqrt{2\pi}} e^{-\frac{1}{2}(x)^2} \int_{\frac{\mu_M+\sigma_M x-\mu_T}{\sigma_T}}^{\infty} \frac{1}{\sqrt{2\pi}} e^{-\frac{1}{2}y^2} dydx}$$
$$= \frac{\int_{\frac{1165-1000}{100}}^{\infty} \frac{1}{\sqrt{2\pi}} e^{-\frac{1}{2}(x)^2} \int_{\frac{1000+100x-1082}{40}}^{\infty} \frac{1}{\sqrt{2\pi}} e^{-\frac{1}{2}y^2} dydx}{\int_{-\infty}^{\infty} \frac{1}{\sqrt{2\pi}} e^{-\frac{1}{2}(x)^2} \int_{\frac{1000+100x-1082}{40}}^{\infty} \frac{1}{\sqrt{2\pi}} e^{-\frac{1}{2}y^2} dydx} = 300 \text{ ppm}.$$

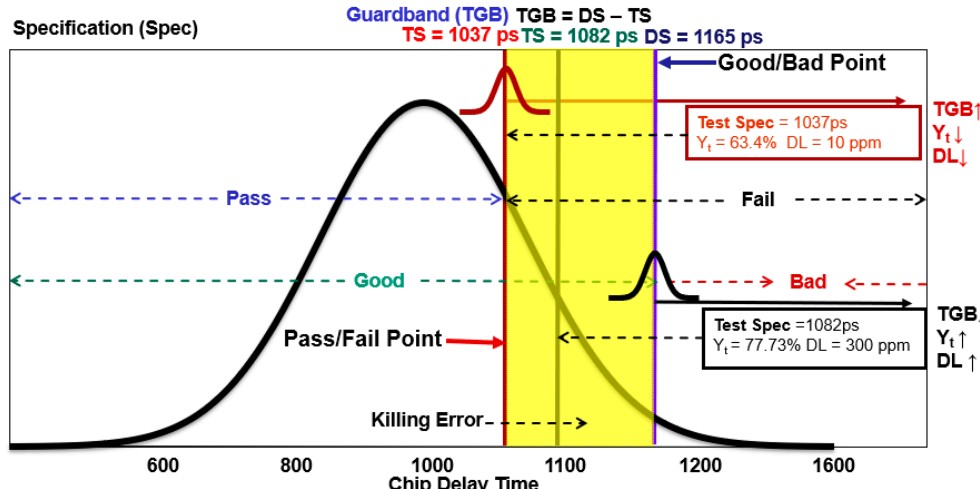

**Figure 6.** Test guardband movement determines the outcome of the test.

The test specification was set to TS = 1037 ps and TGB = 128 ps could be obtained (TGB, 1165 ps − 1037 ps). After estimating the formula above, we can obtain the test yield $(Y_t)$ = 63.4% and perform a high-quality test (DL = 10 ppm).

$$\text{Test Yield}(Y_t)R_{1t}^{1+}$$

$$= \int_{-\infty}^{\infty} \frac{1}{\sigma_M\sqrt{2\pi}}e^{-\frac{1}{2}\left(\frac{X-\mu_M}{\sigma_M}\right)^2} \int_x^{\infty} \frac{1}{\sigma_T\sqrt{2\pi}}e^{-\frac{1}{2}\left(\frac{y-\mu_T}{\sigma_T}\right)^2} dydx$$

$$= \int_{-\infty}^{\infty} \frac{1}{\sigma_M\sqrt{2\pi}}e^{-\frac{1}{2}\left(\frac{X-1000}{100}\right)^2} \int_x^{\infty} \frac{1}{\sigma_T\sqrt{2\pi}}e^{-\frac{1}{2}\left(\frac{y-1037}{40}\right)^2} dydx$$

$$= \int_{-\infty}^{\infty} \frac{1}{\sqrt{2\pi}} e^{-\frac{1}{2}(x)^2} \int_{\frac{\mu_M+\sigma_M x-\mu_T}{\sigma_T}}^{\infty} \frac{1}{\sqrt{2\pi}}e^{-\frac{1}{2}y^2} dydx$$

$$= \int_{-\infty}^{\infty} \frac{1}{\sqrt{2\pi}}e^{-\frac{1}{2}(x)^2} \int_{\frac{1000+100x-1037}{40}}^{\infty} \frac{1}{\sqrt{2\pi}}e^{-\frac{1}{2}y^2} dydx = 63.4\%.$$

DL (Defect Level)

$$= \frac{\int_{\frac{DS-\mu_M}{\sigma_M}}^{\infty} \frac{1}{\sqrt{2\pi}}e^{-\frac{1}{2}(x)^2} \int_{\frac{\mu_M+\sigma_M x-\mu_T}{\sigma_T}}^{\infty} \frac{1}{\sqrt{2\pi}}e^{-\frac{1}{2}y^2} dydx}{\int_{-\infty}^{\infty} \frac{1}{\sqrt{2\pi}}e^{-\frac{1}{2}(x)^2} \int_{\frac{\mu_M+\sigma_M x-\mu_T}{\sigma_T}}^{\infty} \frac{1}{\sqrt{2\pi}}e^{-\frac{1}{2}y^2} dydx}$$

$$= \frac{\int_{\frac{1165-1000}{100}}^{\infty} \frac{1}{\sqrt{2\pi}}e^{-\frac{1}{2}(x)^2} \int_{\frac{1000+100x-1037}{40}}^{\infty} \frac{1}{\sqrt{2\pi}}e^{-\frac{1}{2}y^2} dydx}{\int_{-\infty}^{\infty} \frac{1}{\sqrt{2\pi}}e^{-\frac{1}{2}(x)^2} \int_{\frac{1000+100x-1037}{40}}^{\infty} \frac{1}{\sqrt{2\pi}}e^{-\frac{1}{2}y^2} dydx} = 10 \text{ ppm.}$$

The abovementioned simulation results indicate that expanding the TGB increases the test quality at the expense of test yield wafer products. Therefore, the expansion of the TGB can guarantee the high quality of the shipment. Conversely, a reduction in the TGB increases the test yield and low-quality product output. In other words, the test yield $(Y_t)$ and test quality of the product can be interchanged, but they cannot achieve both. In the foundry's wafer manufacturing process, various uncertain error factors may cause IC product defects. In addition, the probability of missing (β) and killing (α) errors in the product may occur due to the ATE inaccuracy and the unsuitable operation method of the engineer during the test. Therefore, the proper selection of the TGB can ensure the high quality of shipments and reduce the missing (β) and killing (α) errors of chips.

*The Accuracy of Automated Test Equipment Affects the Test Results*

The value of the OTA can be used to indicate the chip tester's accuracy and testing capability [5,6]. A low OTA value (superior accuracy) indicates that the manufacturing capability is inferior to the ATE testing capability, and its price is quite expensive (millions of

dollars). On the contrary, a considerable OTA value indicates that the ATE testing capability is poor and the ability to distinguish chips is also limited. As shown in Figure 7 and Table 1, an IC tester (ATE) with different accuracies was used to test the wafer device under test (DUT). To cite an instance, the company developed a chip with a design specification set to 1165 ps. Design house designed a chip whose electrical characteristic parameters were as follows: $X \sim N$ (x; $\mu_M$ = 1000 ps and $\sigma_M$ = 100 ps). Substituting this into the previously estimated Formula (2) yields a manufacturing yield of $Y_m$ = 95%. First, an IC tester (ATE), where $\sigma_T$ = 40 ps (the lower the value of $\sigma_T$, the higher the accuracy of the ATE), was used to test the DUT, OTA = $\sigma_T \times 3$ = 120 ps. The test specification parameter, TS = 1082 ps, was also utilized to test the DUT, and we could obtain the TGB (1165 ps − 1082 ps = 83 ps). Formula (2) was used to estimate the test yield $Y_t$ = 77.73% and product quality (DL = 300 ppm). In addition, an IC tester (ATE) with low precision, $\sigma_T$ = 60 ps (the higher the value of $\sigma_T$, the lower the accuracy of the ATE, OTA = $\sigma_T \times 3$ = 180 ps), was chosen to test the DUT. Under the same quality condition (DL = 300 ppm), the test specification adopted TS = 1028 ps, and we could obtain a test yield of $Y_t$ = 59.65%. From the abovementioned test results, the test yield was poor (killing and missing error wafers also increased) when using a low-precision tester (OTA). Conversely, using high-precision automated test equipment (OTA) not only increased the test yield, but also maintained a certain product quality level.

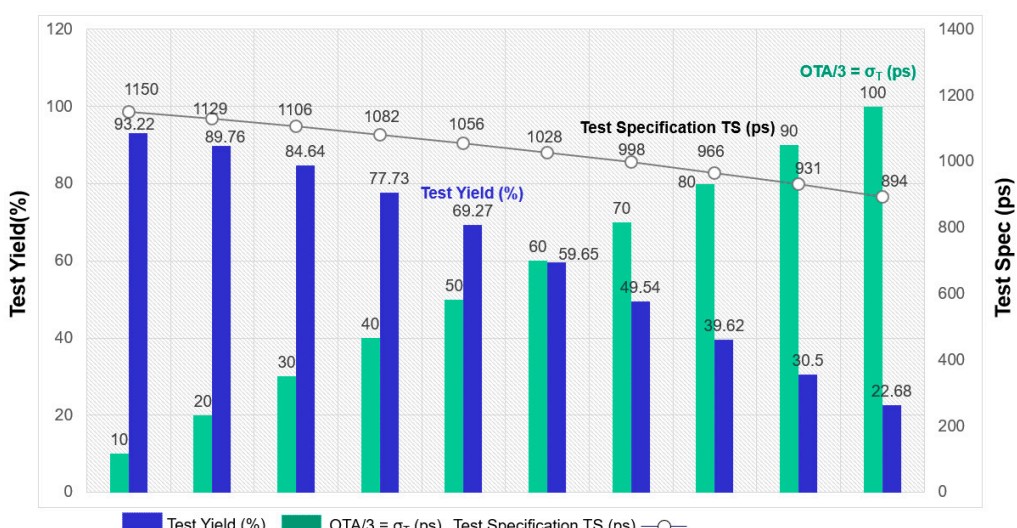

**Figure 7.** Accuracy of the IC tester and the test specifications interact to affect the test results.

**Table 1.** Test TGB impact on test yield.

| OTA/3 = $\sigma_T$ | ps | 100 | 90 | 80 | 70 | 60 | 50 | 40 | 30 | 20 | 10 |
|---|---|---|---|---|---|---|---|---|---|---|---|
| Test specification (TS) | ps | 894 | 931 | 966 | 998 | 1028 | 1056 | 1082 | 1106 | 1129 | 1150 |
| TGB = DS − TS | ps | 271 | 234 | 199 | 167 | 137 | 109 | 83 | 59 | 36 | 15 |
| $Y_t$ | % | 22.68 | 30.5 | 39.62 | 49.54 | 59.65 | 69.27 | 77.73 | 84.64 | 89.76 | 93.22 |
| DL | ppm | 300 | 300 | 300 | 300 | 300 | 300 | 300 | 300 | 300 | 300 |
| $Y_m$ | % | 95% | 95% | 95% | 95% | 95% | 95% | 95% | 95% | 95% | 95% |

The abovementioned simulation results indicate that, with the use of high-precision OTA accuracy testers, the occurrence of missing (β) and killing (α) errors is reduced and an ideal test quality ($Y_q$) and yield ($Y_t$) can be obtained. A high-precision (OTA) IC tester (ATE) is costly for companies. Based on the cost considerations of the company and market demand, choosing a cost-effective and appropriate IC tester is necessary for decision makers.

## 4. Retest Solution to Improve the Yield

At present, the progress of automatic test equipment (ATE) still lags behind the progress of the semiconductor fabrication process. Therefore, using a slow-developing instrument to select electronic products that meet the TSs is important. The current testing technology and capabilities cannot meet the needs of customers; thus, the testing house must find useful testing methods to solve the problem of backward IC tester technology. Therefore, the industry and academia have proposed different methods for retesting schemes [12–18]. For example, Teslence Technology Co., Ltd. (TT) developed an approach to testing and applied it to the production line, which could effectively improve the test yield ($Y_t$) rates of IC products [14]. Based on the abovementioned ideas, the test methods and conditions were adjusted and the test time of the DUT as extended. The DUT of the chip was also tested, considering the reasonable test cost. During the testing process, it was highly important to choose a suitable test point to meet the $Y_t$ and $Y_q$ requirements. First, the quality of chip product testing was considered and the clients set the quality specifications of their products. Then, four test points (TS = DS; TS = DS − $1\sigma_T$; TS = DS − $2\sigma_T$; TS = DS − $3\sigma_T$) were selected to test the DUT, and the test results were estimated. Then, the TGB was appropriately moved based on the estimated DL. Finally, the most suitable test specification, TS, was determined when the product met the quality conditions. The movement of the TGB and different retesting methods were applied to three retesting methods [20–22], including (1) repeat, (2) unbalance (3), and multiple tests.

### 4.1. Scheme 1: Recycling Test Method $M_R^{2+}$

After DUT testing, the TGB can be divided into failed (F) and passed (P) parts. The number of wafers that were killing errors could be classified by the IC tester in the failed (F) part due to IC tester's inaccuracy or improper use of the TGB, accounting for a considerable proportion. Therefore, the test time was extended and the test approach and conditions were modified to retest the failed (F) part such that the chips that passed the second time were tested again for the third time [20]. Figure 8 presents the decision diagram of the recycling test method; the formula for the recycling test ( $M_R^{2+}$ ) method can be defined as follows:

$$
\begin{aligned}
Y_t = Y_P + Y_{FPP} &= M_R^{2+} \text{Test Yield}(\%) \\
&= \int_{-\infty}^{\infty} \text{Chip}(x, \mu_M) \int_x^{\infty} \text{Tester}(y, \mu_T) dy dx \\
&+ \int_{-\infty}^{\infty} \text{Chip}(x, \mu_M) \int_{-\infty}^x \text{Tester}(y, \mu_T) dy \int_x^{\infty} \text{Tester}(z, \mu_T) dz \int_x^{\infty} \text{Tester}(w, \mu_T) dw dx \\
&= \int_{-\infty}^{\infty} \frac{1}{\sigma_M \sqrt{2\pi}} e^{\frac{-(x-\mu_M)^2}{2\sigma_M^2}} \int_x^{\infty} \frac{1}{\sigma_T \sqrt{2\pi}} e^{\frac{-(y-\mu_T)^2}{2\sigma_T^2}} dy dx + \int_{-\infty}^{\infty} \frac{1}{\sigma_M \sqrt{2\pi}} e^{\frac{-(x-\mu_M)^2}{2\sigma_M^2}} \int_{-\infty}^X \frac{1}{\sigma_T \sqrt{2\pi}} e^{\frac{-(y-\mu_T)^2}{2\sigma_T^2}} dy \int_x^{\infty} \frac{1}{\sigma_T \sqrt{2\pi}} e^{\frac{-(z-\mu_T)^2}{2\sigma_T^2}} dz \int_x^{\infty} \frac{1}{\sigma_T \sqrt{2\pi}} e^{\frac{-(w-\mu_T)^2}{2\sigma_T^2}} dw dx \\
&= \int_{-\infty}^{\infty} \frac{1}{\sqrt{2\pi}} e^{-\frac{1}{2}(x)^2} \int_{\frac{\mu_M + \sigma_M x - \mu_T}{\sigma_T}}^{\infty} \frac{1}{\sqrt{2\pi}} e^{-\frac{1}{2}y^2} dy dx + \int_{-\infty}^{\infty} \frac{1}{\sqrt{2\pi}} e^{-\frac{1}{2}(x)^2} \int_{-\infty}^{\frac{\mu_M + \sigma_M x - \mu_T}{\sigma_T}} \frac{1}{\sqrt{2\pi}} e^{-\frac{1}{2}(y)^2} dy \int_{\frac{\mu_M + \sigma_M x - \mu_T}{\sigma_T}}^{\infty} \frac{1}{\sqrt{2\pi}} e^{-\frac{1}{2}z^2} dz \int_{\frac{\mu_M + \sigma_M x - \mu_T}{\sigma_T}}^{\infty} \frac{1}{\sqrt{2\pi}} e^{-\frac{1}{2}w^2} dw dx,
\end{aligned}
$$
(6)

$$
\begin{aligned}
&\text{Defect Level} = DL(\text{ppm}) \\
&= \frac{\text{Missing Errors}}{Y_t} = \frac{\int_{DS}^{\infty} \frac{1}{\sigma_M \sqrt{2\pi}} e^{\frac{-(x-\mu_M)^2}{2\sigma_M^2}} \int_x^{\infty} \frac{1}{\sigma_T \sqrt{2\pi}} e^{\frac{-(y-\mu_T)^2}{2\sigma_T^2}} dy dx + \int_{DS}^{\infty} \frac{1}{\sigma_M \sqrt{2\pi}} e^{\frac{-(x-\mu_M)^2}{2\sigma_M^2}} \int_{-\infty}^x \frac{1}{\sigma_T \sqrt{2\pi}} e^{\frac{-(y-\mu_T)^2}{2\sigma_T^2}} dy \int_x^{\infty} \frac{1}{\sigma_T \sqrt{2\pi}} e^{\frac{-(z-\mu_T)^2}{2\sigma_T^2}} dz \int_x^{\infty} \frac{1}{\sigma_T \sqrt{2\pi}} e^{\frac{-(w-\mu_T)^2}{2\sigma_T^2}} dw dx}{\int_{-\infty}^{\infty} \frac{1}{\sigma_M \sqrt{2\pi}} e^{\frac{-(x-\mu_M)^2}{2\sigma_M^2}} \int_x^{\infty} \frac{1}{\sigma_T \sqrt{2\pi}} e^{\frac{-(y-\mu_T)^2}{2\sigma_T^2}} dy dx + \int_{-\infty}^{\infty} \frac{1}{\sigma_M \sqrt{2\pi}} e^{\frac{-(x-\mu_M)^2}{2\sigma_M^2}} \int_{-\infty}^x \frac{1}{\sigma_T \sqrt{2\pi}} e^{\frac{-(y-\mu_T)^2}{2\sigma_T^2}} dy \int_x^{\infty} \frac{1}{\sigma_T \sqrt{2\pi}} e^{\frac{-(z-\mu_T)^2}{2\sigma_T^2}} dz \int_x^{\infty} \frac{1}{\sigma_T \sqrt{2\pi}} e^{\frac{-(w-\mu_T)^2}{2\sigma_T^2}} dw dx} \\
&= \frac{\int_{\frac{DS-\mu_M}{\sigma_M}}^{\infty} \frac{1}{\sqrt{2\pi}} e^{-\frac{x^2}{2}} \int_{\frac{\mu_M+\sigma_M x-\mu_T}{\sigma_T}}^{\infty} \frac{1}{\sqrt{2\pi}} e^{-\frac{y^2}{2}} yx + \int_{\frac{DS-\mu_M}{\sigma_M}}^{\infty} \frac{1}{\sqrt{2\pi}} e^{-\frac{x^2}{2}} \int_{-\infty}^{\frac{\mu_M+\sigma_M x-\mu_T}{\sigma_T}} \frac{1}{\sqrt{2\pi}} e^{-\frac{y^2}{2}} y \int_{\frac{\mu_M+\sigma_M x-\mu_T}{\sigma_T}}^{\infty} \frac{1}{\sqrt{2\pi}} e^{-\frac{z^2}{2}} z \int_{\frac{\mu_M+\sigma_M x-\mu_T}{\sigma_T}}^{\infty} \frac{1}{\sqrt{2\pi}} e^{-\frac{w^2}{2}} w dx}{\int_{-\infty}^{\infty} \frac{1}{\sqrt{2\pi}} e^{-\frac{1}{2}(x)^2} \int_{\frac{\mu_M+\sigma_M x-\mu_T}{\sigma_T}}^{\infty} \frac{1}{\sqrt{2\pi}} e^{-\frac{1}{2}y^2} dy dx + \int_{-\infty}^{\infty} \frac{1}{\sqrt{2\pi}} e^{-\frac{1}{2}(x)^2} \int_{-\infty}^{\frac{\mu_M+\sigma_M x-\mu_T}{\sigma_T}} \frac{1}{\sqrt{2\pi}} e^{-\frac{1}{2}(y)^2} dy \int_{\frac{\mu_M+\sigma_M x-\mu_T}{\sigma_T}}^{\infty} \frac{1}{\sqrt{2\pi}} e^{-\frac{1}{2}z^2} dz \int_{\frac{\mu_M+\sigma_M x-\mu_T}{\sigma_T}}^{\infty} \frac{1}{\sqrt{2\pi}} e^{-\frac{1}{2}w^2} dw dx}
\end{aligned}
$$
(7)

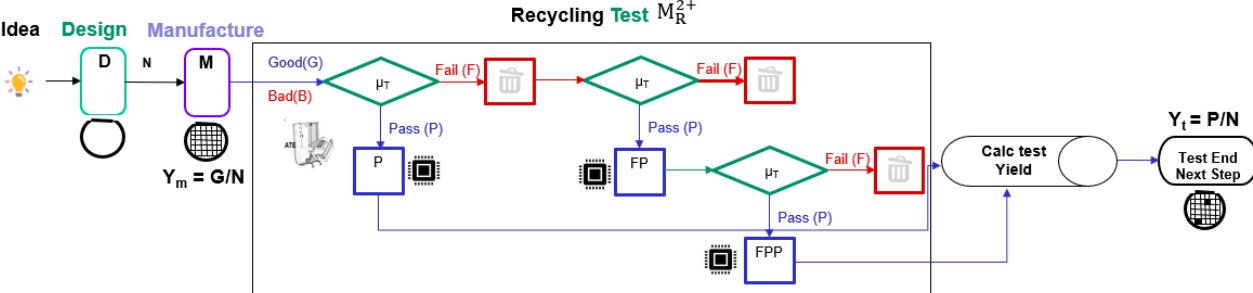

**Figure 8.** Decision diagram for recycling test ( $M_R^{2+}$ ).

*4.2. Scheme 2: Multiple Test Method $M_{2t}^{2+}$*

The retesting test method was adopted and different TSs were used to test the DUT of the part that passed the first retest, which is called the multiple test [21]. The DUT can be divided into two parts after experiencing the first testing process: passed (P) and failed (F). Good parts were selected for retesting and different TSs were used, maintaining the P part. This retest method is called a multiple test (Figure 9). The ( $M_{2t}^{2+}$ ) test result formula is defined below.

$$
\begin{aligned}
\text{Test Yield}(\%)Y_t &= \left(M_{2t}^{2+}\right) \\
&= \int_{-\infty}^{\infty} \text{Chip}(X, \mu_M) \int_x^\infty \text{Tester}\left(y, \mu_{T_1}\right) dy \int_x^\infty \text{Tester}\left(z, \mu_{T_2}\right) dz dx \\
&= \int_{-\infty}^{\infty} \frac{1}{\sigma_M\sqrt{2\pi}} e^{\frac{-(x-\mu_M)^2}{2\sigma_M^2}} \int_x^\infty \frac{1}{\sigma_T\sqrt{2\pi}} e^{\frac{-(y-\mu_{T1})^2}{2\sigma_T^2}} dy \int_x^\infty \frac{1}{\sigma_T\sqrt{2\pi}} e^{\frac{-(z-\mu_{T2})^2}{2\sigma_T^2}} dz dx \\
&= \int_{-\infty}^{\infty} \frac{1}{\sqrt{2\pi}} e^{-\frac{1}{2}(x)^2} \int_{\frac{\mu_M+\sigma_M x-\mu_{T1}}{\sigma_T}}^\infty \frac{1}{\sqrt{2\pi}} e^{-\frac{1}{2}y^2} dy \int_{\frac{\mu_M+\sigma_M x-\mu_{T2}}{\sigma_T}}^\infty \frac{1}{\sqrt{2\pi}} e^{-\frac{1}{2}z^2} dz dx,
\end{aligned}
\tag{8}
$$

$$
\begin{aligned}
&\text{Defect Level} = \text{DL}(\text{ppm}) \\
&= \frac{\text{Missing Errors}}{Y_t} = \frac{\int_{DS}^\infty \frac{1}{\sigma_M\sqrt{2\pi}} e^{-\frac{1}{2}\left(\frac{X-\mu_M}{\sigma_M}\right)^2} \int_x^\infty \frac{1}{\sigma_T\sqrt{2\pi}} e^{-\frac{1}{2}\left(\frac{y-\mu_{T1}}{\sigma_T}\right)^2} dy \int_x^\infty \frac{1}{\sigma_T\sqrt{2\pi}} e^{-\frac{1}{2}\left(\frac{z-\mu_{T2}}{\sigma_T}\right)^2} dz dx}{\int_{-\infty}^\infty \frac{1}{\sigma_M\sqrt{2\pi}} e^{\frac{-(x-\mu_M)^2}{2\sigma_M^2}} \int_x^\infty \frac{1}{\sigma_T\sqrt{2\pi}} e^{\frac{-(y-\mu_{T1})^2}{2\sigma_T^2}} dy \int_x^\infty \frac{1}{\sigma_T\sqrt{2\pi}} e^{\frac{-(z-\mu_{T2})^2}{2\sigma_T^2}} dz dx} \\
&= \frac{\int_{\frac{DS-\mu_M}{\sigma_M}}^\infty \frac{1}{\sqrt{2\pi}} e^{-\frac{x^2}{2}} \int_{\frac{\mu_M+\sigma_M x-\mu_{T1}}{\sigma_T}}^\infty \frac{1}{\sqrt{2\pi}} e^{-\frac{y^2}{2}} y \int_{\frac{\mu_M+\sigma_M x-\mu_{T2}}{\sigma_T}}^\infty \frac{1}{\sqrt{2\pi}} e^{-\frac{z^2}{2}} dz\mathbf{dx}}{\int_{-\infty}^\infty \frac{1}{\sqrt{2\pi}} e^{-\frac{x^2}{2}} \int_{\frac{\mu_M+\sigma_M x-\mu_{T1}}{\sigma_T}}^\infty \frac{1}{\sqrt{2\pi}} e^{-\frac{y^2}{2}} dy \int_{\frac{\mu_M+\sigma_M x-\mu_{T2}}{\sigma_T}}^\infty \frac{1}{\sqrt{2\pi}} e^{-\frac{z^2}{2}} dz dx}.
\end{aligned}
\tag{9}
$$

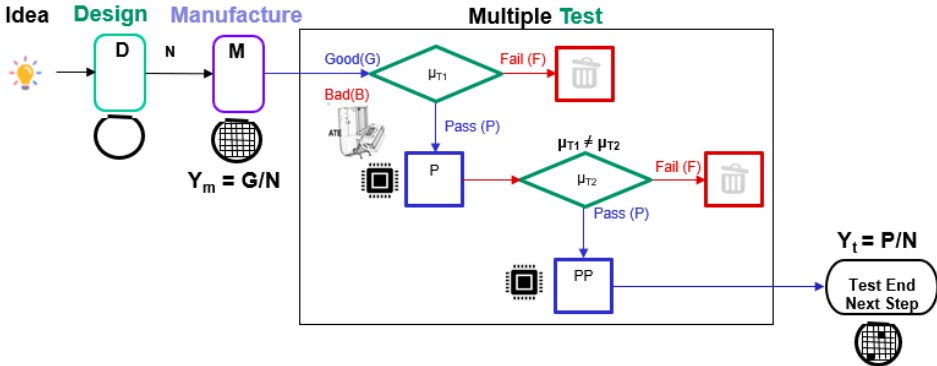

**Figure 9.** Decision diagram for multiple test ( $M_{2t}^{2+}$ ).

*4.3. Scheme 3: Repeat Test Method $R_{2t}^{2+}$*

The test conditions and methods were modified, the number of tests was increased, and the same TS was used to test the DUTs that passed the first retest [22], which is called

repeated testing. The multiple-test decision-making process is shown in Figure 10. The first test was performed and the chip DUT was divided into failed (F) an passed (P) parts after the test. The (P) part included the missing error part; thus, retesting was conducted through the (P) DUT part and the same TSs were used to enhance the yield ($Y_t$) and test quality ($Y_q$). This retesting decision-making method is called the multiple test, and the symbol is expressed as $R_{2t}^{2+}$. The following is the test yield formula for repeat testing:

$$\begin{aligned}
Y_t = \text{Test Yield}(\%) &= \left(R_{2t}^{2+}\right)\\
&= \int_{-\infty}^{\infty} \text{Chip}(X, \mu_M) \int_x^{\infty} \text{Tester}(y, \mu_T) dy \int_x^{\infty} \text{Tester}(z, \mu_T) dz dx\\
&= \int_{-\infty}^{\infty} \frac{1}{\sigma_M \sqrt{2\pi}} e^{\frac{-(x-\mu_M)^2}{2\sigma_M^2}} \int_x^{\infty} \frac{1}{\sigma_T \sqrt{2\pi}} e^{\frac{-(y-\mu_T)^2}{2\sigma_T^2}} dy \int_x^{\infty} \frac{1}{\sigma_T \sqrt{2\pi}} e^{\frac{-(z-\mu_T)^2}{2\sigma_T^2}} dz dx\\
&= \int_{-\infty}^{\infty} \frac{1}{\sqrt{2\pi}} e^{-\frac{1}{2}(x)^2} \int_{\frac{\mu_M+\sigma_M x-\mu_T}{\sigma_T}}^{\infty} \frac{1}{\sqrt{2\pi}} e^{-\frac{1}{2}y^2} dy \int_{\frac{\mu_M+\sigma_M x-\mu_T}{\sigma_T}}^{\infty} \frac{1}{\sqrt{2\pi}} e^{-\frac{1}{2}z^2} dz dx,
\end{aligned} \tag{10}$$

$$\begin{aligned}
&DL(\text{ppm}) = \text{Defect Level}\\
&= \frac{\text{Missing Error}}{Y_t} = \frac{\int_{DS}^{\infty} \frac{1}{\sigma_M \sqrt{2\pi}} e^{-\frac{1}{2}\left(\frac{X-\mu_M}{\sigma_M}\right)^2} \int_x^{\infty} \frac{1}{\sigma_T \sqrt{2\pi}} e^{-\frac{1}{2}\left(\frac{y-\mu_T}{\sigma_T}\right)^2} dy \int_x^{\infty} \frac{1}{\sigma_T \sqrt{2\pi}} e^{-\frac{1}{2}\left(\frac{z-\mu_T}{\sigma_T}\right)^2} dz dx}{\int_{-\infty}^{\infty} \frac{1}{\sigma_M \sqrt{2\pi}} e^{\frac{-(x-\mu_M)^2}{2\sigma_M^2}} \int_x^{\infty} \frac{1}{\sigma_T \sqrt{2\pi}} e^{\frac{-(y-\mu_T)^2}{2\sigma_T^2}} dy \int_x^{\infty} \frac{1}{\sigma_T \sqrt{2\pi}} e^{\frac{-(z-\mu_T)^2}{2\sigma_T^2}} dz dx}\\
&= \frac{\int_{\frac{DS-\mu_M}{\sigma_M}}^{\infty} \frac{1}{\sqrt{2\pi}} e^{-\frac{x^2}{2}} \int_{\frac{\mu_M+\sigma_M x-\mu_T}{\sigma_T}}^{\infty} \frac{1}{\sqrt{2\pi}} e^{-\frac{y^2}{2}} y \int_{\frac{\mu_M+\sigma_M x-\mu_T}{\sigma_T}}^{\infty} \frac{1}{\sqrt{2\pi}} e^{-\frac{z^2}{2}} z x}{\int_{-\infty}^{\infty} \frac{1}{\sqrt{2\pi}} e^{-\frac{x^2}{2}} \int_{\frac{\mu_M+\sigma_M x-\mu_T}{\sigma_T}}^{\infty} \frac{1}{\sqrt{2\pi}} e^{-\frac{y^2}{2}} dy \int_{\frac{\mu_M+\sigma_M x-\mu_T}{\sigma_T}}^{\infty} \frac{1}{\sqrt{2\pi}} e^{-\frac{z^2}{2}} dz dx}.
\end{aligned} \tag{11}$$

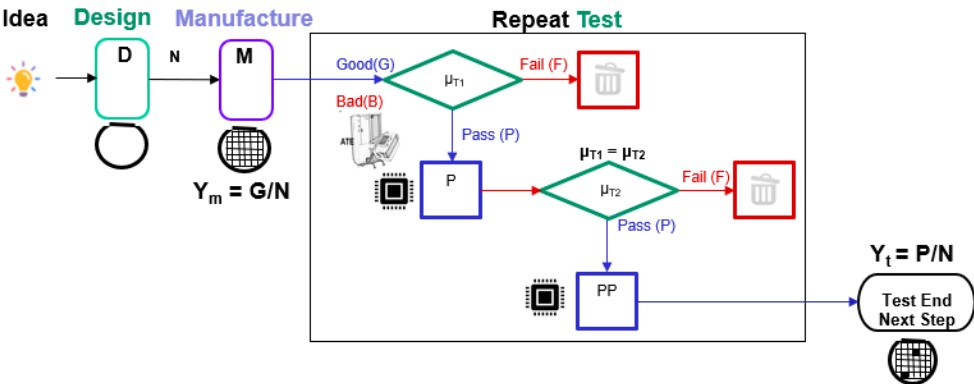

**Figure 10.** Decision diagram for repeat testing ($R_{2t}^{2+}$).

## 5. Apply Three Yield Improvement Test Solutions to the 2021 IRDS Datasheet (300 ppm)

According to the ITRS (International Technology Roadmap for Semiconductors) roadmap [1–3], the speed of progress for the wafers' testing capabilities was different from that of manufacturing technology for semiconductor development. Facing the rapid progress of manufacturing technology and stagnant testing technology, the distinction between good and bad chip circuits in the future will become an important issue. Therefore, if no remarkable technological breakthrough in the progress of process technology relative to the progress of future test methods and IC testers (ATEs) emerges, then the test results (yield rate and quality) will continue to worsen. Moreover, additional efforts should be made to address future processes for advanced chips because IC testers (ATEs) fail to determine good or bad chip circuits. Therefore, under the consideration of ensuring the high quality of wafer products, the test method of retesting ((1) repeat, (2) unbalance, and (3) multiple tests) is proposed to overthrow the concept of quality for yield and yield for quality. The TGB was used to retest the failed wafers to repeatedly find reliable products.

First, the traditional testing method, $R_{1t}^{1+}$, was used to the test protocol (Table 20) described in the 2021 IRDS [23] (shown in Table 2 and Figure 11). The chip developed in 2023 has a DS = 294 ps and the electrical parameter characteristics of the DUT are N(x; 204 ps, 70 ps); thus, the manufacturing yield is $Y_m$ = 95%. Then, the testing capabilities of future IC testers may deteriorate considering the slow progress of the IC tester. Therefore, the data for IRDS 2021 were used as a reference and the ATE tester (OTA = 75 ps) was widely employed in the current operations to test the wafers manufactured for the future (2019–2033). The required test quality (DL = 300 ppm) was maintained during this time and the DUT was tested with a wafer tester where OTA = 75 ps. Meanwhile, a test yield of 67.2% could be obtained by using the $R_{1t}^{1+}$ method and setting the test specification (TS) to 237 ps. Then, the chips produced in 2033 can be tested. The DUT electrical parameters were N(x; 158 ps, 54 ps) and the chip design specification was DS = 227 ps. The test chip was tested using an ATE with an OTA = 75 ps, while maintaining a 300 ppm DL quality. Setting the test point at 166 ps provided a test yield of 55.3%. As shown in Figure 11, when the chip is tested using the $R_{1t}^{1+}$ method, the test yield gradually deteriorates due to the inaccuracy of the tester.

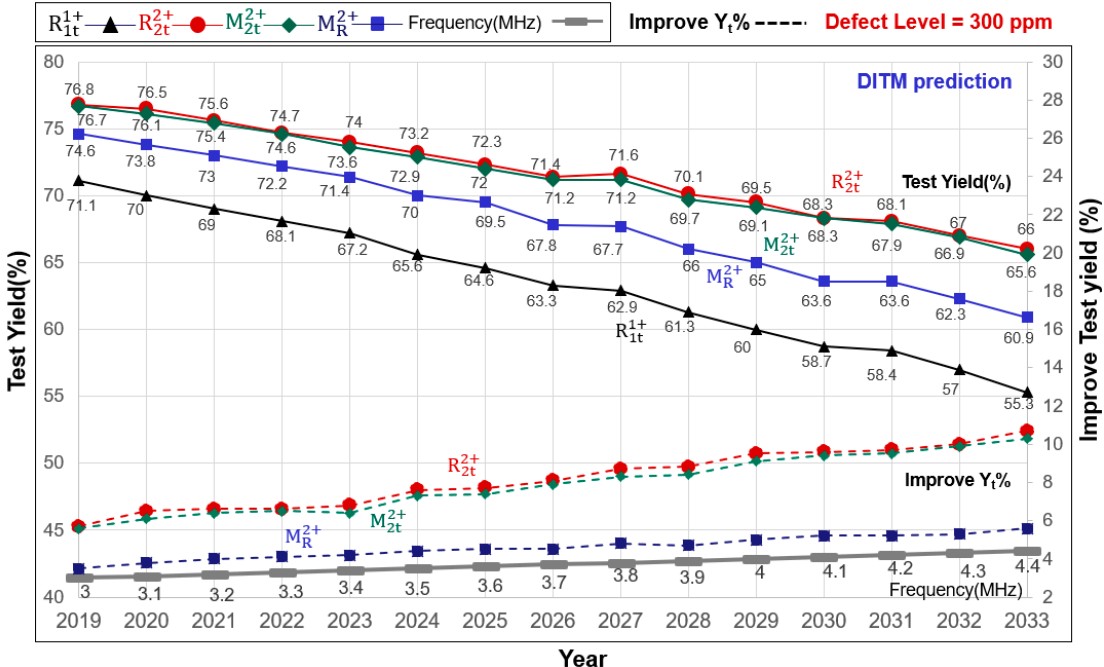

**Figure 11.** Retesting test method improves the test capability of ATE (300 ppm).

The GRC (Group for Reliable Computing) team proposed three effective retesting schemes to improve the test yield ($Y_t$). First, the recycling test method, $M_R^{2+}$, in scheme 1 was used to test the wafers in 2023 under the same defect-level (DL = 300 ppm) conditions. When we set $\mu_T$ to 238 ps, the yield ($Y_t$) of the test could be increased to 71.4% ($Y_t$ could improve: 71.4% − 67.2% = 4.2%). The recycling test method, $M_R^{2+}$, improved the test yield by adjusting the TGB and retesting the wrong part twice. Then, when the test time cost was not considered, two consecutive tests were conducted using the scheme-2 multiple test method, $M_{2t}^{2+}$. When TS $\mu_{T1}$ was 268 ps and $\mu_{T2}$ was 261 ps, the test yield, $Y_t$, could be increased by 6.4% (73.6% − 67.2% = 6.4%). The optimization results of the test method show improvements in the test yield ($Y_t$) by reducing the killing and missing errors and obtaining a certain test quality.

**Table 2.** Application of the retesting method to the IRDS 2021 table (300 ppm).

| Year | | Unit | 2019 | 2020 | 2021 | 2022 | 2023 | 2024 | 2025 | 2026 | 2027 | 2028 | 2029 | 2030 | 2031 | 2032 | 2033 |
|---|---|---|---|---|---|---|---|---|---|---|---|---|---|---|---|---|---|
| Device period | | us | 0.33 | 0.32 | 0.31 | 0.30 | 0.29 | 0.28 | 0.27 | 0.27 | 0.26 | 0.25 | 0.25 | 0.24 | 0.23 | 0.233 | 0.22 |
| Chip frequency | | GHz | 3.0 | 3.1 | 3.2 | 3.3 | 3.4 | 3.5 | 3.6 | 3.7 | 3.8 | 3.9 | 4.0 | 4.1 | 4.2 | 4.3 | 4.4 |
| $\sigma_M$ | | ps | 79 | 76 | 74 | 72 | 70 | 68 | 66 | 64 | 62 | 61 | 59 | 58 | 56 | 55 | 54 |
| $\mu_M$ | | ps | 231 | 223 | 217 | 210 | 204 | 199 | 193 | 188 | 182 | 178 | 174 | 170 | 165 | 162 | 158 |
| OTA | | ps | 75 | 75 | 75 | 75 | 75 | 75 | 75 | 75 | 75 | 75 | 75 | 75 | 75 | 75 | 75 |
| DL | | ppm | 300.0 | 300.0 | 300.0 | 300.0 | 300.0 | 300.0 | 300.0 | 300.0 | 300.0 | 300.0 | 300.0 | 300.0 | 300.0 | 300.0 | 300.0 |
| $R_{1t}^{1+}$ | $Y_t$ | % | 71.1 | 70 | 69 | 68.1 | 67.2 | 65.6 | 64.6 | 63.3 | 62.9 | 61.3 | 60 | 58.7 | 58.4 | 57 | 55.3 |
| | $TS(\mu_T)$ | ps | 277 | 265 | 256 | 246 | 237 | 228 | 220 | 211 | 204 | 197 | 190 | 184 | 178 | 173 | 166 |
| $M_R^{2+}$ | $Y_t$ | % | 74.6 | 73.8 | 73 | 72.2 | 71.4 | 70 | 69.5 | 67.8 | 67.7 | 66 | 65 | 63.6 | 63.6 | 62.3 | 60.9 |
| $TS(\mu_T)$ | | ps | 278 | 266 | 257 | 247 | 238 | 229 | 221 | 212 | 205 | 197 | 192 | 185 | 178 | 173 | 167 |
| Yield↑ | Improvement | % | 3.5 | 3.8 | 4 | 4.1 | 4.2 | 4.4 | 4.5 | 4.5 | 4.8 | 4.7 | 5 | 5.2 | 5.2 | 5.3 | 5.6 |
| $M_{2t}^{2+}$ | $Y_t$ | % | 76.7 | 76.1 | 75.4 | 74.6 | 73.6 | 72.9 | 72 | 71.2 | 71.2 | 69.7 | 69.1 | 68.3 | 67.9 | 66.9 | 65.6 |
| TS1($\mu_{T1}$) | $\mu_T$ | ps | 308 | 297 | 287 | 277 | 268 | 259 | 251 | 243 | 237 | 229 | 223 | 216 | 210 | 205 | 199 |
| TS2($\mu_{T2}$) | | | 302 | 291 | 282 | 271 | 261 | 254 | 246 | 237 | 230 | 314 | 217 | 211 | 204 | 199 | 192 |
| Yield↑ | Improvement | % | 5.6 | 6.1 | 6.4 | 6.5 | 6.4 | 7.3 | 7.4 | 7.9 | 8.3 | 8.4 | 9.1 | 9.4 | 9.5 | 9.9 | 10.3 |
| $R_{2t}^{2+}$ | $Y_t$ | % | 76.8 | 76.5 | 75.6 | 74.7 | 74 | 73.2 | 72.3 | 71.4 | 71.6 | 70.1 | 69.5 | 68.3 | 68.1 | 67 | 66 |
| $TS(\mu_T)$ | | ps | 305 | 294 | 284 | 274 | 265 | 257 | 248 | 240 | 234 | 226 | 220 | 214 | 207 | 202 | 196 |
| Yield↑ | Improvement | % | 5.7 | 6.5 | 6.6 | 6.6 | 6.8 | 7.6 | 7.7 | 8.1 | 8.7 | 8.8 | 9.5 | 9.6 | 9.7 | 10 | 10.7 |

The scheme-3 repeat test method, $R_{2t}^{2+}$, was adopted and TS was selected to test the semiconduction chip ($\mu_T$ = 265 ps). The yield ($Y_t$) of the test could be improved from 67.2% to 74%% ($Y_t$ could improve the value: 74% − 67.2% = 6.8%). The estimated results reveal that all three retesting methods can effectively improve the test yield. Compared with the traditional test method, under identical defect-level condition (DL = 300 ppm) requirements, the retest method can increase the test yield by approximately 3.5% to 10.7%. In addition, a comparison of the estimated results for the three retest methods shows that using the repeat test method, $R_{2t}^{2+}$, can maximize the yield ($Y_t$) of the test without sacrificing the test quality ($Y_q$).

The abovementioned comparison results reveal that the application of the retesting scheme to the product testing of general quality (300 ppm) has the following advantages:

(1) The operation is simple; the test guardband is moved and the test method is changed.
(2) Missing ($\beta$) and killing ($\alpha$) errors are reduced.
(3) The yield ($Y_t$) of the test is increased.
(4) The capability of the ATE substantially improves.
(5) The chips that can be sold increase.
(6) The profits of the company increase.

### 5.1. Use Retesting Test Method to Select High-Quality (10 ppm) Good Chips

The global sales of electric vehicles continue to increase, accounting for a increasing proportion of overall passenger vehicle sales. At present, the scale of the electric vehicle electronics market continues to expand, and the application scope of high-quality chips is also increasing. The design functions of electric vehicles include automatic driving, automatic parking, automatic braking, and collision warning. With the diversification of electronic equipment used in electric vehicles, the number of vehicle chip components continues to increase. The issues of safety and reliability have become important due to the increasing number of IC components. In addition, zero-defect IC chips have become the goal of high-end automotive electronics, and the testing house has invested resources to improve automatic test equipment (ATE) and propose effective verification and test methods. For example, the AEC adopts the AEC-Q001 [9] specification and applies the PAT method to eliminate problematic parts to enhance the quality of IC components. Therefore, the GRC team proposed three effective retesting schemes by adjusting the TGB and changing the TSs, and adjusting the TGB to reduce the inaccuracy of the automated test equipment, improve the test quality and test yield ($Y_t$), and achieve the ultimate goal of high-quality electronic products with zero defects.

Similarly, we referred to the estimated chip electrical data of IRDS 2021, under the premise of maintaining high-quality products (DL = 10 ppm); an IC tester (ATE) where OTA = 75 ps and the traditional test method, $R_{1t}^{1+}$, were utilized to test the wafers produced in 2023 (Figure 12 and Table 3). The TS $\mu_T$ of the traditional test method, $R_{1t}^{1+}$, was set as 210 ps and a test yield of 53.2% was obtained. Then, the recycling test method, $M_R^{2+}$, of scheme 1 was employed to test the wafer in 2023. TS $\mu_T$ = 211 ps was chosen and the test yield rate was 57.2% ($Y_t$ increased: 57.2% − 53.2% = 4%). Then, when the test time was extended, the multiple test method, $M_{2t}^{2+}$, of scheme 2 was adopted to test the chips to be tested thrice. At this point, setting the TS $\mu_{T1}$ = 246 ps and $\mu_{T2}$ = 241 ps, the resulting yield ($Y_t$) could be increased by 10.5% (63.7% − 53.2% = 10.5%). Finally, the yield ($Y_t$) of the test could be improved from 53.2% to 63.9% by using the repeat test method, $R_{2t}^{2+}$, of scheme 3 and the test specification of 244 ps to test the DUT ($Y_t$ increased: 63.9% − 53.2% = 10.7%). The four aforementioned test methods were compared to obtain high-quality zero-defect products (DL = 10 ppm). The abovementioned estimation results show that the yield rate obtained by the three retesting schemes is much better than that the $R_{1t}^{1+}$ method. Particularly, the yield rate enhanced by the repeat testing of scheme three was better than other schemes, that is, the retesting test method can effectively improve the capability of the IC tester (ATE) and markedly reduce the occurrence of missing ($\beta$) and killing ($\alpha$) errors. Furthermore, the retesting test method can improve inaccuracy errors and reduce the occurrence of errors in the IC tester (ATE).

**Table 3.** Application of the retesting method to the IRDS 2021 table (10 ppm).

| Year | | Unit | 2019 | 2020 | 2021 | 2022 | 2023 | 2024 | 2025 | 2026 | 2027 | 2028 | 2029 | 2030 | 2031 | 2032 | 2033 |
|---|---|---|---|---|---|---|---|---|---|---|---|---|---|---|---|---|---|
| Device period | | us | 0.33 | 0.32 | 0.31 | 0.30 | 0.29 | 0.28 | 0.27 | 0.27 | 0.26 | 0.25 | 0.25 | 0.24 | 0.23 | 0.233 | 0.22 |
| Chip frequency | | GHz | 3.0 | 3.1 | 3.2 | 3.3 | 3.4 | 3.5 | 3.6 | 3.7 | 3.8 | 3.9 | 4.0 | 4.1 | 4.2 | 4.3 | 4.4 |
| $\sigma_M$ | | ps | 79 | 76 | 74 | 72 | 70 | 68 | 66 | 64 | 62 | 61 | 59 | 58 | 56 | 55 | 54 |
| $\mu_M$ | | ps | 231 | 223 | 217 | 210 | 204 | 199 | 193 | 188 | 182 | 178 | 174 | 170 | 165 | 162 | 158 |
| OTA | | ps | 75 | 75 | 75 | 75 | 75 | 75 | 75 | 75 | 75 | 75 | 75 | 75 | 75 | 75 | 75 |
| DL | | ppm | 10.0 | 10.0 | 10.0 | 10.0 | 10.0 | 10.0 | 10.0 | 10.0 | 10.0 | 10.0 | 10.0 | 10.0 | 10.0 | 10.0 | 10.0 |
| $R_{1t}^{1+}$ | $Y_t$ | % | 59 | 57.9 | 56.6 | 54.7 | 53.2 | 50.6 | 50 | 48.3 | 47.6 | 45.2 | 43.8 | 41.8 | 41 | 39.6 | 37.5 |
| | $TS(\mu_T)$ | ps | 250 | 239 | 230 | 219 | 210 | 200 | 193 | 185 | 178 | 170 | 164 | 157 | 151 | 146 | 139 |
| $M_R^{2+}$ | $Y_t$ | % | 62.5 | 61.5 | 60.3 | 58.6 | 57.2 | 54.7 | 54.6 | 52.4 | 51.8 | 49.4 | 48 | 46.1 | 45.3 | 44 | 42.4 |
| $TS(\mu_T)$ | | ps | 250 | 239 | 230 | 220 | 211 | 201 | 194 | 185 | 178 | 170 | 164 | 158 | 151 | 146 | 139 |
| Yield↑ | Improvement | % | 3.5 | 3.6 | 3.7 | 3.9 | 4 | 4.1 | 4.1 | 4.1 | 4.2 | 4.2 | 4.2 | 4.3 | 4.3 | 4.4 | 4.9 |
| $M_{2t}^{2+}$ | $Y_t$ | % | 68.4 | 67.8 | 66.4 | 65.3 | 63.7 | 62.6 | 62 | 60.3 | 60.1 | 57.9 | 56.8 | 55.6 | 55.1 | 53.8 | 51.9 |
| TS1($\mu_{T1}$) | $\mu_T$ | ps | 286 | 275 | 266 | 256 | 246 | 238 | 230 | 222 | 215 | 207 | 201 | 195 | 189 | 184 | 177 |
| TS2($\mu_{T2}$) | | | 283 | 271 | 262 | 252 | 241 | 234 | 225 | 219 | 211 | 203 | 197 | 191 | 185 | 180 | 173 |
| Yield↑ | Improvement | % | 9.4 | 9.9 | 9.8 | 10.6 | 10.5 | 12 | 12 | 12 | 12.5 | 12.7 | 13 | 13.8 | 14.1 | 14.2 | 14.4 |
| $R_{2t}^{2+}$ | $Y_t$ | % | 68.5 | 68 | 66.6 | 65.5 | 63.9 | 62.7 | 62 | 60.5 | 60.2 | 58 | 57 | 55.8 | 55.3 | 54 | 52 |
| $TS(\mu_T)$ | $\mu_T$ | ps | 284 | 273 | 264 | 254 | 244 | 236 | 228 | 220 | 213 | 205 | 199 | 193 | 187 | 182 | 175 |
| Yield↑ | Improvement | % | 9.5 | 10.1 | 10 | 10.8 | 10.7 | 12.1 | 12 | 12.2 | 12.6 | 12.8 | 13.2 | 14 | 14.3 | 14.4 | 14.5 |

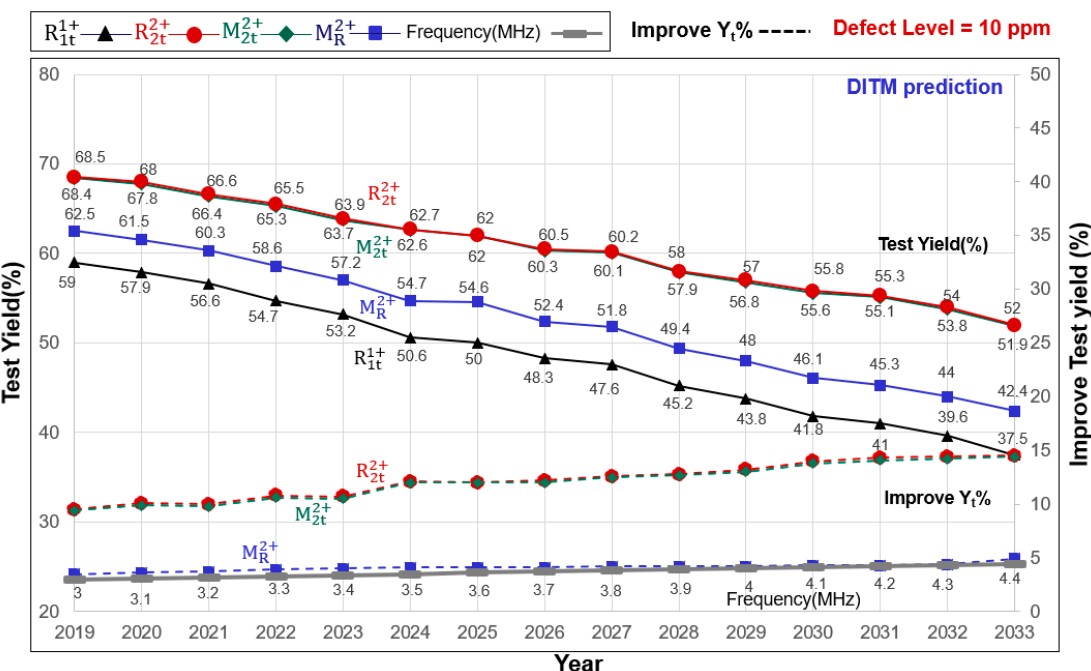

**Figure 12.** Retesting test method improves the test capability of the IC tester (10 ppm).

By contrast, the retest method, $R_{2t}^{2+}$, can increase the yield ($Y_t$) of the test the most, which is 10% higher than the $R_{1t}^{1+}$ method. Affected by many factors, the global shortage of automotive chips has forced automakers to significantly reduce their production capacity. Using the retesting method to recycle the incorrectly manufactured wafers not only solves the chip shortage problem but also creates additional profits for the company and increases its brand value. The comparison results indicate that the application of the retesting scheme for high-quality product testing (10 ppm) demonstrates the following advantages:

(1)   The operation is simple; it only involves adjusting the TGB and changing the test method.
(2)   The proposed scheme can markedly reduce killing and missing errors.
(3)   It can substantially improve the yield ($Y_t$) of the test.
(4)   It can substantially enhance the ability of the ATE.
(5)   The proposed scheme can significantly increase the number of chips that can be sold.
(6)   More high-quality chips can be selected.
(7)   The profits of the company significantly increase.

## 5.2. Retesting Scheme Advantages

When the repeated testing method was used to test the DUT and the testing cost exceeded the profit through the increase in yield, this method lost its practical value. Since each wafer varied in complexity and performance, the cost of testing followed the same pattern. If the cost of wafers was considered, the calculation of the finished products and sales became increasingly complicated. In addition, as the number of tests increased, the evaluation of the maximum profit that the increased test cost and yield that could achieved became more complex. Therefore, a detailed analysis, calculation, and estimation of the overall market were required to obtain correct and reliable data. Multiple $M_{2t}^{2+}$ and repeated $R_{2t}^{2+}$ test methods reduced the probability of the α error by increasing the TGB. The β errors could be reduced and $Y_q$ could be improved by increasing the number of tests and adjusting the test specifications. Moreover, multiple tests not only improve $Y_t$, but also maintain $Y_q$ at a certain level. Comparing the various methods presented above [12–18], the three retesting schemes proposed by our GRC team offer the following advantages:

(1) The proposed solution does not require spending considerable time collecting large wafer data and can reduce software development costs.

(2) This solution can be based on the estimated data, and the trend curve of the future wafer, $Y_t$, can be calculated.

(3) No additional hardware equipment is required, which not only reduces the cost of testing but also controls the relative quality of the product.

(4) DITM's rapid calculation is used to estimate the yield trend of the product. Primarily based on the effective data (specifications of the components proposed by the manufacturer) and the model of the testing machine (instrument parameters), the required test specifications can be rapidly calculated.

(5) They are achieved by adjusting the TGB to avoid $\alpha$ errors and reduce $\beta$ errors, which increases $Y_t$ and improves $Y_q$.

(6) They can effectively improve the performance of the testing machine.

(7) They can use the current testing machines to screen out high-quality wafers.

## 6. Conclusions

A DITM model, which could efficiently analyze the influence of test parameters and semiconductor processes on $Y_q$ and $Y_t$, was proposed. According to the estimates reported by the ITRS, the speed of improvement of testing capabilities failed to match the capabilities of semiconductor manufacturing processes. If the developments of IC tester (ATE) capabilities and test methods have no additional breakthroughs in the future, $Y_t$ will deteriorate over time. In addition, in the past two years, the impact of the new crown epidemic and the increasing demand for chips for new energy vehicles has caused a shortage of automotive chips in the global semiconductor industry. Thus, the shortage of chips will become a serious issue in the future. Therefore, each testing factory actively explores valid test methods to improve the problem of an inadequate test capacity.

Therefore, our Group for Reliable Computing (GRC) team used valid data (specifications of components proposed by the manufacturers) and an IC tester (instrument parameters) and could rapidly calculate $Y_t$ through the DITM's calculation methods. Without needing additional hardware equipment, a retest test plan was proposed. Furthermore, under the condition of setting fixed test quality parameters, the number and duration of tests increased. By adjusting the TGB, we could avoid killing ($\alpha$) errors and reduce missing ($\beta$) errors, as well as achieve the goal of improving the overall yield. Not only can $Y_t$ be increased, but $Y_q$ can also be maintained at a certain level. Concurrently, it can also reduce the time and cost of the test. Based on the retest plan, the GRC team developed three retesting schemes to improve $Y_t$ and $Y_q$ to meet the demands of consumers for high-quality products. The three retesting methods relied on the adjustment of the TGB to increase $Y_t$ and reduce $\beta$ and $\alpha$ errors. Moreover, the three proposed testing schemes were demonstrated by using a set of IRDS 2021 parameters, which enhanced $Y_t$ and maintained $Y_q$. The estimated results presented above reveal that the retesting scheme improves the test ability of the ATE and enhances $Y_t$. Simultaneously, this scheme increased the sales of high-quality chips and alleviated the serious shortage of automotive chips worldwide.

**Author Contributions:** Methodology, C.-H.Y. and J.-E.C.; Software, C.-H.Y.; Validation, C.-H.Y.; Formal analysis, C.-H.Y. and J.-E.C.; Investigation, C.-H.Y.; Resources, C.-H.Y.; Data curation, C.-H.Y.; Writing—original draft, C.-H.Y.; Writing—review & editing, C.-H.Y.; Visualization, C.-H.Y.; Supervision, C.-H.Y.; Project administration, C.-H.Y.; Funding acquisition, C.-H.Y. All authors have read and agreed to the published version of the manuscript.

**Funding:** This research received no external funding.

**Institutional Review Board Statement:** Not applicable.

**Informed Consent Statement:** Not applicable.

**Data Availability Statement:** All data are included within manuscript.

**Conflicts of Interest:** The authors declare no conflict of interest.

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
