# Peer review of "Retesting Schemes That Improve Test Quality and Yield Using a Test Guardband"

_2673-4117, doi:10.3390/eng4040169_

Round 1

Reviewer 1 Report

Comments and Suggestions for Authors

In this manuscript, C.H. Yeh and J.E. Chen proposed retesting schemes that improve test quality and yield through test guardband. They tried three schemes: recycling testing method, multiple test method, and repeat test method, then found out that the 3rd scheme repeat test method can increase the yield of the test the most, which is 10% higher than R1+_1t.

The main question addressed by this paper is the testing accuracy in semiconductor industry. Testing can either create killing error or missing error, which is important issue in semiconductor industry. This manuscript discussed potential solutions theoretically and used IEEE database to test the improvement of the proposed methods. The mathematical derivation is very helpful to the general readers, but more discussion is needed such as comparison with other methods in other publications.

I recommend major revision before it can be published on Eng. My comments are as follows:

1.       This manuscript needs 3rd party language improvement. It has many grammatical typos.

2.       The authors may need to refer to scientific paper writing structure so that a clear problem and solution statement are given in the Introduction section. Conclusion section needs to be the summary of the methodology and results, instead of introduction. The current manuscript seems to be taken from a thesis chapter. It needs to be rewritten into a scientific paper.

3.       Table2 is a little hard to read due to too much information. The authors may want to break that into sub tables for each method.  

4.       The reference format is not consistent. The authors need to add more relevant references such as in no.5 and fix the format.

5.       The authors also need to compare the improvement of other methods versus the three methods in this paper, such as 

Z. Jiang and S. K. Gupta, ”Threshold Testing: Improving Yield for Nanoscale VLSI,” in IEEE Transactions on Computer-Aided Design of Integrated Circuits and Systems, vol. 28, no. 12, pp. 1883-1895, Dec. 2009.

K. Lee, T. Hsieh and M. A. Breuer, ”Efficient Overdetection Elimination of Acceptable Faults for Yield Improvement,” in IEEE Transactions on Computer-Aided Design of Integrated Circuits and Systems, vol. 31, no. 5, pp. 754-764, May 2012

Comments on the Quality of English Language

3rd party English improvment service is required

Author Response

Dear Sir,

Please find, in the submission section of the authors, our final response to the comments received from the two reviewers to

" Retesting Schemes that Improve Test Quality and Yield through Test Guardband."

We appreciate your comments very much, as they have pointed out a number of issues that need to be addressed. We would like to thank the editor and the reviewers for taking time reading and suggesting modifications to the paper, and your cogent comments have proven to be very useful for the improvement of the paper. We did several modifications to the initial manuscript based on the suggestions of the reviewers. We hope that the editor will find the paper eligible for publication. In the answers we have explicate all the changes we have done. We hope that this will be useful for the new review. Thank you very much for your kind consideration of this resubmitted version of our manuscript.

Sincerely yours,

Chung-Huang Yeh,

(On behalf of the authors of the manuscript)

Reviewer 2 Report

Comments and Suggestions for Authors

It is a good paper that analyzes the degree of field improvement by analyzing the recycling test, multiple test, and repeated test, and is recognized as a technology to increase productivity in the future.

<Modifications Requested for the Paper>

1. Add the proposed technology and the results of the proposed technology to the Abstract.

2. These abbreviations, such as DS, TS, and GRC, need to be defined before use.

<Additional comments requested>

1. Need to further explain the meaning of expressions such as N(x; 158 ps, 54 ps) below the 359 line.

1. In below lines 171 and 196

1-1) About standard deviation value

- Need to add more reason set to 100 ps.

- Need to add what 100 ps means.

1-2) About the OTA

- Need to add more reason for setting to 120ps.

- Need to add what 120ps means.

- Can you show the OTA in Figure 4 for ease of understanding.

2. In Table 1, if the TGB decreases, Yt should increase, but there is a contradiction that Yt decreases. Need further clarification to help reader understand.

3. The Recycling test, which verifies the Fail DUT again, increases Yt, but the Multiple test and repeated test, which test the pass DUT once more, requires a decrease in Yt, but in your results, it is increasing mostly. Additional explanation of why.

4. Although the advantages of increasing Yt for the three types of test methods have been well explained, it is also necessary to mention the evaluation of each type of test methods as the test cost considering the test time increases.

5. Need to further explain from what perspective TGB can be adjusted in the 3 test approaches, and need to explain TGB adjustment effect.

Author Response

(The authors gave the same response as above.)

Round 2

Reviewer 1 Report

Comments and Suggestions for Authors

The authors have addressed all my concerns 

Author Response

Dear Sir,

Thank you for your comments. 

Sincerely yours,

Chung-Huang Yeh, PhD

Reviewer 2 Report

Comments and Suggestions for Authors

Most of the comments were revised and reflected. It would just be nice to add an explanation by numerically showing the degree to which the proposed technique has been improved effectively in the abstract.

Author Response

Dear Sir,

Please find, in the submission section of the authors, our final response to the comments received from the two reviewers to

" Retesting Schemes that Improve Test Quality and Yield through Test Guardband."

We appreciate your comments very much, as they have pointed out a number of issues that need to be addressed. We would like to thank the editor and the reviewers for taking time reading and suggesting modifications to the paper, and your cogent comments have proven to be very useful for the improvement of the paper. We did several modifications to the initial manuscript based on the suggestions of the reviewers. We hope that the editor will find the paper eligible for publication. In the answers we have explicate all the changes we have done. We hope that this will be useful for the new review. Thank you very much for your kind consideration of this resubmitted version of our manuscript.

Sincerely yours,

Chung-Huang Yeh,
